# The Role of Zinc in Gliotoxin Biosynthesis of *Aspergillus fumigatus*

**DOI:** 10.3390/ijms20246192

**Published:** 2019-12-08

**Authors:** Hyewon Seo, Suzie Kang, Yong-Sung Park, Cheol-Won Yun

**Affiliations:** School of Life Sciences and Biotechnology, Korea University, Anam-dong, Sungbuk-gu, Seoul 02841, Korea; hyewon330@korea.ac.kr (H.S.); sthe327@korea.ac.kr (S.K.); dcomtrue@korea.ac.kr (Y.-S.P.)

**Keywords:** Zinc, gliotoxin, *ZafA*, *GliZ*, *A. fumigatus*

## Abstract

Zinc performs diverse physiological functions, and virtually all living organisms require zinc as an essential trace element. To identify the detailed function of zinc in fungal pathogenicity, we carried out cDNA microarray analysis using the model system of *Aspergillus fumigatus*, a fungal pathogen. From microarray analysis, we found that the genes involved in gliotoxin biosynthesis were upregulated when zinc was depleted, and the microarray data were confirmed by northern blot analysis. In particular, zinc deficiency upregulated the expression of *GliZ*, which encodes a Zn2-Cys6 binuclear transcription factor that regulates the expression of the genes required for gliotoxin biosynthesis. The production of gliotoxin was decreased in a manner inversely proportional to the zinc concentration, and the same result was investigated in the absence of *ZafA*, which is a zinc-dependent transcription activator. Interestingly, we found two conserved ZafA-binding motifs, 5′-CAAGGT-3′, in the upstream region of *GliZ* on the genome and discovered that deletion of the *ZafA*-binding motifs resulted in loss of ZafA-binding activity; gliotoxin production was decreased dramatically, as demonstrated with a *GliZ* deletion mutant. Furthermore, mutation of the ZafA-binding motifs resulted in an increase in the conidial killing activity of human macrophage and neutrophil cells, and virulence was decreased in a murine model. Finally, transcriptomic analysis revealed that the expression of *ZafA* and *GliZ* was upregulated during phagocytosis by macrophages. Taken together, these results suggest that zinc plays an important role in the pathogenicity of *A. fumigatus* by regulating gliotoxin production during the phagocytosis pathway to overcome the host defense system.

## 1. Introduction

Iron, copper, and zinc are representative essential trace elements and perform many physiological roles in living organisms. The function of iron, copper, and zinc in living cells has been studied in various model organisms. Iron is an essential metal ion that works as a cofactor of many enzymes, and iron deficiency results in fatal diseases, such as hereditary anemia [1]. On the other hand, iron overload induces the so-called Fenton reaction, which mediates the production of hydroxyl radicals and is fatal to living cells by affecting DNA and proteins [2]. For this reason, iron homeostasis is strictly regulated. Copper also carries out many important roles in living organisms, and copper deficiency or overload causes severe genetic diseases, such as Menkes disease or Wilson disease [3], respectively. Zinc deficiency also causes several diseases, such as impaired immune function, growth defects, and psychological disorders in humans [4,5,6]. Iron, copper, and zinc are divalent metal ions that carry out oxidoreduction reactions and produce reactive oxygen species that affect cellular activity.

These three metals also have important functions in the pathogenesis of microbial pathogens, and most of the gene products involved in metal metabolism work as virulence factors [7]. Loss of the uptake systems for these metals in microbial pathogens results in loss of virulence, and this phenomenon is observed in many different microbial pathogens, such as *Candida albicans* [8,9], *Cryptococcus neoformans* [10,11], *Aspergillus fumigatus* [12,13], and *Neisseria* spp. [14,15]. When microbial pathogens infect a host, competition for metal acquisition between the host and pathogens occurs. For example, when infecting a host, *A. fumigatus* is confronted with macrophage phagocytosis, and zinc is depleted by the host zinc transporter system during the phagocytic pathway [16]. Microbial pathogens cannot use zinc from the environment and fail to grow in macrophages [17]. Therefore, we tried to further identify the detailed role of zinc in the pathogenicity of *A. fumigatus*.

Zinc exists as a variety of forms and acts as a cofactor in many proteins, such as transcriptional factors, metallothioneins, and various zinc-binding proteins [18,19]. *A. fumigatus*, which is a fungal pathogen, needs zinc to carry out physiological functions. Zinc uptake by *A. fumigatus* depends on the ZIP (Zrt-, Irt-like proteins) family encoded by *ZrfA*, *ZrfB*, and *ZrfC* [20]. Zinc transporters efficiently take up zinc in zinc-limiting conditions from the extracellular environment into the cytoplasm. ZrfC functions as a major zinc transporter because of its unusually long N-terminus, whereas ZrfA and ZrfB seem to play an accessory role [12,21]. Interestingly, the expression of *ZrfC* is higher in alkaline conditions than in acidic environments, while the *ZrfA* and *ZrfB* expression patterns are opposite to that of *ZrfC* [22]. ZafA is a zinc-responsive transcriptional activator and is an ortholog of the *Saccharomyces cerevisiae* Zap1 transcription factor [23]. In acidic conditions, ZafA binds to the promoter regions of *ZrfA* and *ZrfB*, whereas in alkaline media, ZafA binds to the *ZrfC* promoter region and activates the expression of those genes [21,24].

Recently, it has been reported that metals control the production of secondary metabolites of fungal species, which act as virulence factors of microbial pathogens [25]. Secondary metabolites are produced by microorganisms, and they perform a protective function against the host [26]. For example, *A. fumigatus* produces several secondary metabolites that are most likely involved in invasive aspergillosis [25]. The transcriptional regulation of secondary metabolite biosynthesis has been studied in *A. fumigatus*, and LaeA, which is a transcriptional activator, has an important function in secondary metabolite production, such as that of melanin, ergot alkaloids, and gliotoxin [27,28,29]. The genes involved in secondary metabolites are clustered in the genome, and gene expression is regulated by LaeA [29]. Gliotoxin is one of the secondary metabolites produced by *A. fumigatus*. Gliotoxin is a nonribosomal peptide and a member of the epipolythiodioxopiperazine (ETP) class of redox-active secondary metabolites naturally produced by *A. fumigatus* and is one of its virulence factors. It contains a disulfide bond bridge across a piperazine ring that seems to be responsible for its toxicity [30,31]. In *A. fumigatus*, gliotoxin is encoded by a cluster composed of 13 genes, namely, *GliZ*, *GliI*, *GliJ*, *GliP*, *GliC*, *GliM*, *GliG*, *GliK*, *GliA*, *GliN*, *GliF*, *GliT*, and a separate *GtmA* gene. It is well known that gliotoxin has multiple immunosuppressive activities and inhibits the nuclear transcription factor NF-kB, especially inducing host cell apoptosis [32,33]. It has also been demonstrated that gliotoxin can inhibit macrophage functions, including phagocytosis [34].

Furthermore, it has been reported that the production of gliotoxin is regulated by zinc availability [35] and that the gene expression of *GliZ* is upregulated by zinc starvation; however, the detailed regulatory mechanism remains to be solved [35]. In this paper, we demonstrate the detailed regulatory mechanism of gliotoxin biosynthesis by zinc.

## 2. Results

### 2.1. Genes Involved in Gliotoxin Production are Downregulated by Zinc

To understand the regulation of global gene expression by zinc in *A. fumigatus*, cDNA microarray analysis was performed. The cells were cultured in AMM-zinc plus 1 µM zinc and standard AMM to exponential phase, and total RNA was extracted. Standard AMM contains 77 µM zinc in the medium. We found that many genes involved in cellular metabolism were down- or upregulated by zinc. Among them, we found that the genes involved in gliotoxin biosynthesis were upregulated under zinc-deficient conditions. As shown in Figure 1, *GliZ*, *GliA*, *GliT*, and *GtmA* were upregulated under zinc-deficient conditions, and other genes were not detected in this experiment because of low expression levels. To confirm the cDNA microarray data, northern blot analysis was performed. The cells were cultured in AMM-zinc medium with 0 to 2 µM zinc, and total RNA was extracted. As shown in Figure 1B, the expression of *GliZ*, *GliA*, *GliT*, and *GtmA* was downregulated in a manner inversely proportional to the zinc concentration, and these results coincided with the microarray data. GliZ is a Zn2-Cys6 binuclear transcription factor that positively regulates gene expression involved in gliotoxin biosynthesis. To investigate the regulatory mechanism of *GliZ* gene expression by zinc, a *ZafA* deletion mutant was constructed, and the expression of the genes involved in gliotoxin biosynthesis was investigated. As shown in Figure 2A, the *ZafA* deletion mutant showed a growth defect in low-zinc and standard AMM media, and supplementation with zinc rescued the growth defect of the *ΔzafA* strain. Additionally, the introduction of *ZafA* into the *ΔzafA* strain rescued the growth defect of the *ΔzafA* strain. As shown in Figure 2B, northern blot analysis was performed with *ΔzafA* and *ZafA*-complemented strains. Each strain was cultured until the exponential phase, and total RNA was extracted. The gene expression of *GliZ*, *GliT, GliA*, and other *gli* cluster genes were decreased dramatically when *ZafA* was deleted, and the introduction of *ZafA* to the *ΔzafA* strain rescued the expression of *GliZ*, *GliT, GliA*, and other *gli* cluster genes. Furthermore, *GliM*, *GliA*, *GliT* and *GliA* gene expression was not detected in the *GliZ* deletion strain (Figure 2C). These data indicate that zinc regulates the expression of the genes involved in gliotoxin biosynthesis by regulating *GliZ* expression.

To further investigate the regulatory mechanism of zinc on gliotoxin biosynthesis, gliotoxin production was measured from the cells of the wild type, *ZafA* deletion strain, and *ZafA*-complemented strain. As shown in Figure 3A, gliotoxin production was measured using HPLC, and extremely low production of gliotoxin was observed from the *ZafA* deletion strain (Appendix A). However, the *ZafA*-complemented strain showed even higher gliotoxin production than the wild-type cells. As shown in Figure 2B, the *ZafA*-complemented strain showed higher expression of *gli* cluster genes and gliotoxin production than the wild-type cells, and these results are thought to be due to the presence of the *pyrG* as the selectable marker of the *ZafA*-complemented strain and activating the expression of target gene. To investigate the involvement of zinc in gliotoxin biosynthesis, gliotoxin production was measured from cells cultured with different zinc concentrations. As shown in Figure 3B, the cells were cultured with the indicated zinc concentration, and the produced gliotoxin was measured (Appendix A). Interestingly, gliotoxin production was decreased in a manner inversely proportional to the zinc concentration, and this result coincided with the result of northern blot analysis, as shown in Figure 1B. The gliotoxin production from the *GliZ* deletion strain was measured (Appendix A), as shown in Figure 3C.

### 2.2. ZafA Binds to the Upstream Region of GliZ and Regulates Gliotoxin Biosynthesis

To identify the mechanism of ZafA regulation of *GliZ* expression, we looked for the conserved ZafA-binding motif [23] in the upstream region of the *GliZ* gene. Interestingly, we found two conserved ZafA-binding motifs, 5′-CAAGGT-3′, from the −850 and −761 positions of the *GliZ* upstream region. To confirm the binding of ZafA to the ZafA-binding motifs of *GliZ*, mutagenesis was performed to remove the 5′-CAAGGT-3′ ZafA-binding motif from the upstream region of *GliZ*, as shown in Figure 4A, and a mutant strain was constructed. The −831-position mutant is M1, the −785-position mutant is M2, and the strain with mutations at both the −831 and −785 positions is M1M2. Mutagenesis had no effect on growth, as shown in Figure 4B. To investigate the effect of ZafA on *GliZ* expression in vitro, an ONPG assay was performed using a beta-galactosidase-fused system. Each upstream region of *GliZ* wild type and each mutant, M1, M2, and M1M2, was fused with the beta-galactosidase gene and subcloned into the yeast vector pRS425. The constructed vectors were cotransformed into yeast cells with a *ZafA* plasmid that expressed ZafA protein in yeast cells. Then, beta-galactosidase activity was measured. As shown in Figure 4C, although the wild-type upstream region of *GliZ* showed high beta-galactosidase activity when *ZafA* was also expressed, the M1M2 mutant showed the lowest beta-galactosidase activity. These results indicate that ZafA binds specifically to the *GliZ* upstream region. To investigate the regulation of *GliZ* by ZafA, northern blot analysis was performed with the mutant strains M1, M2, and M1M2. As shown in Figure 4D, the gene expression levels of *GliZ*, *GliT*, and *GliA* were downregulated in the M1M2 mutant, and this result showed the function of ZafA in gliotoxin biosynthesis.

To further confirm the function of ZafA in *GliZ* expression, an EMSA was performed. ZafA protein was purified using a His-tagged form of ZafA from *E. coli*, and overexpressed recombinant ZafA was used in the EMSA. As shown in Figure 4E, the wild-type ZafA-binding motif and each mutant of *GliZ* were amplified and labeled with the ^32^P radioisotope. The ZafA protein bound strongly to the wild-type ZafA-binding motif, and a mobility shift was observed. An unlabeled wild-type ZafA-binding motif competed with the labeled ZafA-binding motif, and the addition of 350-fold excess of the cold ZafA-binding motif completely inhibited the mobility shift. However, the M1 and M2 mutants partially failed to compete with the ZafA-binding motif, and M1M2 completely failed to compete with the ZafA-binding motif. Furthermore, we measured gliotoxin production using the M1 and M2 mutants (Appendix A). As shown in Figure 4F, the indicated strains were cultured as described in the materials and methods section, and gliotoxin production was measured. Gliotoxin production was decreased when the ZafA-binding motif was mutated, and the M1M2 strain showed the lowest gliotoxin production, which coincided with the northern blot analyses, EMSAs, and ONPG assays. These results indicate that ZafA binds specifically to the ZafA-binding motifs of *GliZ* and regulates gene expression of *GliZ* and other *gli* cluster genes as well as ultimately regulating gliotoxin production.

### 2.3. ZafA Binding to the ZafA-Binding Motif of GliZ is a Virulence Factor

Gliotoxin is a virulence factor, and strains defective in gliotoxin biosynthesis partially fail to infect hosts [36]. To identify the involvement of ZafA binding to the ZafA-binding motifs of *GliZ* in the pathogenesis of *A. fumigatus*, we performed a conidial killing assay with human macrophage and neutrophil cell lines. As shown in Figure 5, wild-type cells and mutant cells were infected with the indicated cell lines. The HL-60 differentiated macrophage cell line [37] was infected with wild-type and mutant strains, and a conidial killing assay was performed. M1 and M2 showed lower viability than wild type, and M1M2 showed the lowest viability among them. The same result was observed in the HL-60 differentiated neutrophil cell line [38]. To further confirm virulence defects, a murine virulence assay was performed. As shown in Figure 5C, wild-type cells killed the mouse very quickly, and all mice died after 6 days of infection. However, M1M2 showed a substantially delayed death rate for which all mice died 12 days after fungal infection. The *GliZ* deletion mutant showed a death rate similar to that of the M1M2 mutant. To investigate fungal lung infection, a fungal burden assay was performed. As shown in Figure 5D, the negative control, which was not infected by *A. fumigatus*, showed clear morphology in hematoxylin and eosin (H&E) and Grocott methenamine silver (GMS) staining, indicating that no fungal infection was present. However, GMS staining of the positive control, which was infected with wild-type cells, showed many fungal cells. In addition, the M1M2 and *GliZ* mutants did not show significant fungal cells from H&E and GMS staining and were similar to the negative control. These results indicate that ZafA plays an important role in gliotoxin biosynthesis by regulating *GliZ* expression and that it is a virulence factor in fungal infection.

### 2.4. Transcriptome Analysis Showed that the Expression of Gli Cluster Genes was Upregulated during Fungal Infection of Macrophages by Activating ZafA Gene Expression

It has been reported that zinc depletion occurs during pathogen infection of macrophages, and this phenomenon is a host defense system against microbial pathogens. To identify the involvement of ZafA in gliotoxin production during fungal infection of macrophages, RNA-seq analysis was performed using phagocytic cells of macrophages with *A. fumigatus*. After infection of macrophages with *A. fumigatus* for 1 and 2 h, total RNA was extracted from the infected cells and RNA-seq analysis was performed. Zero infection time was the comparative control. As shown in Figure 6A, transcriptome analysis revealed that the expression of genes involved in metal metabolism and secondary metabolite biosynthesis was regulated and that the genes involved in zinc metabolism were upregulated specifically after infection. Interestingly, we found that *gli* cluster genes were also upregulated. To confirm the RNA-seq data, qRT-PCR was performed. As shown in Figure 6B, *ZafA* and three zinc transporters, *ZrtA*, *ZrtB*, and *ZrtC*, were upregulated after infection with *A. fumigatus*. *GliZ* and other *gli* cluster genes were also upregulated after infection. These data coincide with our previous in vitro data and indicate that zinc regulates the pathogenicity of *A. fumigatus* during phagocytosis by regulating gliotoxin biosynthesis. Finally, we constructed transcriptome network modeling of the genes involved in zinc, copper, iron, and gliotoxin metabolism during the phagocytosis pathway. As shown in Figure 6C, the genes involved in zinc, copper, iron, and gliotoxin metabolism were grouped individually, and up- and downregulated genes are shown. From the transcriptome network, we found that *ZafA* and *GliZ* were upregulated during phagocytosis and that copper- and iron-related genes were also upregulated. Here, we focused on the relationship between *ZafA* and *GliZ*. As shown in Figure 6C, the genes involved in zinc metabolism, such as *ZafA* and *ZrfC*, were upregulated. Furthermore, the genes involved in gliotoxin biosynthesis, such as GliZ and GliT, were upregulated. We hypothesized from our results that zinc starvation during phagocytosis upregulated *ZafA* first and then *GliZ*. Finally, gliotoxin biosynthesis was upregulated even when zinc depletion was caused by host cells. These data indicate that zinc regulates gliotoxin biosynthesis directly and that the involvement of other metals in secondary metabolite biosynthesis should be identified further.

## 3. Discussion

Metals have diverse physiological functions in living organisms, and their deficiency or abundance causes fatal diseases in humans. Therefore, metal homeostasis is strictly regulated in living organisms. Furthermore, metals play an important role in the pathogenicity of microbial pathogens, and the utilization of metals such as iron, copper, and zinc is a known virulence factor [7]. *A. fumigatus* is a saprophytic fungus that causes aspergillosis in humans. *A. fumigatus* has both a siderophore-mediated and a reductive iron uptake system for iron acquisition [39], and dysfunction of the siderophore-mediated iron uptake system results in failure to infect host cells [13]. Deletion of the gene encoding *SidA*, which catalyzes siderophore biosynthesis, results in loss of virulence [40]. Copper utilization ability is also a virulence factor in *A. fumigatus*. Our group identified the copper transporters, Ctr proteins, and the copper-dependent transcription activator AfMac1, and the deletion of *Afmac1* resulted in growth defects in low-copper concentrations and partial loss of virulence [41,42]. Furthermore, ZafA is a zinc-dependent transcription activator of *A. fumigatus*, and the deletion of *ZafA* also results in growth defects and loss of virulence [23]. These reports indicate that iron, copper, and zinc play important roles in the pathogenicity of microbial pathogens although their detailed mechanisms have not yet been identified, and elucidation of the detailed function of metals in pathogenicity is required.

Previously, we and other groups reported that Afmac1 regulates expression of the genes involved in several secondary metabolites, including siderophores, gli-like genes, and the metabolites of pyoverdine biosynthesis [43]. Furthermore, it has been reported recently that PpzA, which is involved in iron assimilation, regulates secondary metabolite biosynthesis, such as that of pyripyropene, TAFC, and fumagillin, under iron starvation conditions [44]. Our data in this report show that ZafA regulates the gene expression of secondary metabolite-encoding genes, especially *gli* cluster genes. These reports indicate that metals regulate the gene expression of secondary metabolites. Secondary metabolites are produced by most microorganisms and carry out diverse functions, such as protective functions against other living organisms, metal transport, differentiation, immunosuppression, and communication [45]. In fungi, secondary metabolites, such as siderophores, antibiotics, and mycotoxins, have been studied. Generally, the genes encoding secondary metabolites in fungi are clustered in the genome, and genes for gliotoxin biosynthesis are also clustered in the genome [30]. Some fungal secondary metabolites have been recognized as virulence factors [46], and gliotoxin is also recognized as a virulence factor of *A. fumigatus* [36]. Gliotoxin has been known as a virulence factor that causes invasive aspergillosis [47], although the function of gliotoxin during pathogenesis remains to be solved. Gliotoxin can produce reactive oxygen species because of its disulfide structure [30], and it serves an immunosuppressive function by inactivating the host defense system [48]. This immunosuppressive function of gliotoxin is involved in many different human diseases. As described previously, we and other groups reported that trace metals regulate secondary metabolite biosynthesis, but the reason why metals affect secondary metabolite production is not yet known. It has been reported that the expression of *gli* cluster genes is regulated by gliotoxin, abiotic factors, and the LaeA transcription factor. Gliotoxin biosynthesis is regulated by gliotoxin itself because none of the synthesized gliotoxins are secreted. The remaining cellular gliotoxin regulates the gene expression of *GliZ*, which is a transcription activator of *gli* cluster genes [49]. Abiotic factors, such as pH, temperature, and aeration, of culture conditions affect the expression of *gli* cluster genes [50,51,52]. LaeA, which is a transcription factor, regulates many secondary metabolites, and gliotoxin is one of them [29]. However, the detailed regulatory mechanism of gliotoxin gene expression has not yet been identified because these factors indirectly regulate the expression of *gli* cluster genes.

Our question is how trace metals regulate gliotoxin biosynthesis. As described previously, gliotoxin has immunosuppressive function, and this function may help *A. fumigatus* infect host cells [32]. Therefore, gliotoxin activity during phagocytosis plays an important role in overcoming the host defense system. When pathogens infect hosts, macrophages attack microbial pathogens by phagocytosis, and diverse host defense systems are engaged during phagocytosis [53]. Metal limitation in macrophage cells is one of the host defense systems [54]. Host cells pump metals outside of macrophages so that pathogens cannot use the metals. Metal limitation inhibits metal utilization by pathogens and inhibits the growth of pathogens [54,55], ultimately resulting in a loss of virulence. In fact, effluxes of iron, copper, and zinc are observed in many phagocytic cells, and the expression of genes encoding metal transporters is upregulated during phagocytosis [55]. On the other hand, microbial pathogens compete with the host defense system to obtain metals from the environment. *A. fumigatus* upregulates the genes involved in the metal uptake system, and our results showed the upregulation of the genes encoding zinc transporters. In addition, upregulation of *gli* cluster genes during the phagocytosis pathway was observed. These data explain why zinc is involved in gliotoxin biosynthesis. Furthermore, it has been reported that zinc is necessary for gliotoxin biosynthesis and that zinc is used as a cofactor of GliJ, which encodes a dipeptidase, during gliotoxin biosynthesis [56].

From our results, we hypothesize a relationship between metal and gliotoxin production during fungal infection. We identified the direct relationship between zinc and gliotoxin biosynthesis shown in Figure 4, in which ZafA binds to the *GliZ* upstream region directly and regulates gliotoxin biosynthesis. This result provides data regarding the relationship between metals and secondary metabolites. We analyzed the gene expression pattern during phagocytosis, and the genes involved in zinc metabolism and gliotoxin biosynthesis were upregulated. This phenomenon results from metal restriction by the host and one of the pathogen defense systems. The expression profiles of the genes involved in zinc metabolism and gliotoxin biosynthesis are summarized in Figure 6C, and we further identified a relationship between *ZafA* and *GliZ*. When pathogens infect the host, zinc limitation occurs inside the host cells, and pathogens upregulate the genes involved in zinc metabolism. ZafA is upregulated and then upregulates the genes for gliotoxin biosynthesis. Finally, gliotoxin plays an important role in the pathogenesis pathway. The relationship between zinc and gliotoxin is linked by the zinc-responsive transcription factor ZafA. Our data showed the function of metals in regulating secondary metabolite biosynthesis and suggests a possibility for effectively carrying out pathogen control by controlling metal metabolism.

## 4. Materials and Methods

### 4.1. Strains, Media and Growth Conditions

For this study, A1160 *A. fumigatus* was used as a wild-type strain, as well as its derivatives. *A. fumigatus* was cultured at 37 °C in *Aspergillus* minimal medium (AMM) (glucose, salt mix, MgSO_4_, Hunter’s trace element (TE) solution) or complete medium (CM; glucose, yeast extract, casamino acid, vitamin solution, salt mix, MgSO_4_, TE solution) [57]. The detailed compositions of these media are listed in Appendix A. All media and added solutions except the 200× MgSO_4_ solution were sterilized by autoclaving at 121 °C for 15 min. As a precipitate is formed by autoclaving, the MgSO4 solution was sterilized by a filter system. A *ZafA* (1g10080) deletion strain (*ΔzafA*) was constructed by transformation of a deletion cassette containing the 5′-UTR and 3′-UTR of the gene and the *PyrG* marker. A *ZafA* overexpression strain (*PthiA. ZafA*) was constructed by inserting the thiamine-responsive *ThiA* promoter (*PthiA*) of *Aspergillus oryzae* into the upstream region of *ZafA* [58]. All *A. fumigatus* derivatives were confirmed by PCR and southern blot. All primer sets used in this study are listed in Appendix A.

### 4.2. Microarray

For microarray analysis, wild-type, *ΔzafA*, and *ZafA*-complemented cells were cultured for 24 h and harvested. Total RNA was extracted using TRIzol^®^ (Invitrogen, Seoul, Korea) from wild-type, *ΔzafA*, and *ZafA*-complemented cells following the manufacturer’s instructions. Microarray platform, labeling, and hybridization for control and test RNAs, the synthesis of target cRNA probes, and hybridization were performed using Agilent’s Low Input Quick Amp WT labeling kit, Two Colors (Agilent Technology, Seoul, Korea) according to the manufacturer’s instructions. Briefly, 0.2 µg of total RNA was mixed with the WT primer mix and incubated at 65 °C for 10 min. cDNA master mix (5× First Strand buffer, 0.1 M DTT, 10 mM dNTP mix, RNase-Out, and MMLV-RT) was prepared and added to the reaction mix. The samples were incubated at 40 °C for 2 h, and then reverse transcription and dsDNA synthesis were terminated by incubating at 70 °C for 15 min. The transcription master mix was prepared according to the manufacturer’s protocol (4× transcription buffer, 0.1 M DTT, NTP mix, 50% PEG, RNase-Out, inorganic pyrophosphatase, T7-RNA polymerase, and cyanine 3/5-CTP). Transcription of dsDNA was performed by adding the transcription master mix to the dsDNA reaction samples and incubating at 40 °C for 2 h. Amplified and labeled cRNA was purified on an RNase mini column (Qiagen, Seoul, Korea) according to the manufacturer’s protocols. The labeled cRNA target was quantified using an ND-1000 spectrophotometer (NanoDrop Technologies, Inc., Wilmington, DE, USA). After checking the labeling efficiency, cyanine 3′-labeled and cyanine 5′-labeled cRNA targets were combined and fragmented by adding 10× blocking agent and 25× fragmentation buffer and incubating at 60 °C for 30 min. The fragmented cRNA was resuspended with 2× hybridization buffer and directly pipetted onto an assembled microarray (MYcroarray.com, Michigan, USA). The arrays were hybridized at 57 °C for 17 h using an Agilent hybridization oven (Agilent Technology, Seoul, Korea). The hybridized microarray was washed according to the manufacturer’s washing protocols (Agilent Technology, Seoul, Korea). Data analysis of the hybridization images was visualized with an Axon GenePix 4000B scanner (Axon Instrument, California, USA), and data quantification was performed using GenePix Pro 6.0 (Axon Instrument, California, USA). The average fluorescence intensity for each spot was calculated, and the local background was subtracted. LOESS normalization was performed using GenoWiz 4.0 (Ocimumbiosolutions, Telangana, India).

### 4.3. Northern Blot

We performed northern blotting as described by Sambrook and Russell [59]. When we performed electrophoresis, we loaded 5 or 10 μg of total RNA on a 1% agarose gel containing formaldehyde. Then, the gel was transferred to a nylon membrane (Hybond-N+ nylon membrane, GE Healthcare). DNA fragments amplified from the internal regions of the indicated genes by PCR were used as probes and labeled with radioisotopes using a random priming kit (GE Healthcare, Seoul, Korea). Hybridization was performed at 65 °C overnight. rRNA was used as a loading control.

### 4.4. Plate Assay

Conidia of each strain were inoculated on AMM agar plates, cultured for 72 h and harvested with a 0.01% Tween 80 solution. The number of spores was counted using a hemocytometer. Then, 0.01% Tween 80 solutions containing 5 × 10^3^, 5 × 10^2,^ and 5 × 10^1^ spores were spotted on AMM agar plates and cultured for 72 h. After autoclaving the AMM without zinc, zinc sulfate solution was added to the media and mixed well.

### 4.5. Site-Directed Mutagenesis

To generate the *GliZ*(6g09630) promoter mutant strains M1, M2, and M1M2, a site-directed mutagenesis kit (Intronbio) was used. A 1-kb *GliZ* genomic DNA sequence was PCR amplified with the *GliZ Xho*I_F and *Hind*III_R primers and then ligated into the pGEM-T Easy vector. This plasmid was used as a template. Primers were 24 bp in length and designed without the 5′-CAAGGT-3′ conserved ZafA-binding motif. The primers used in this mutagenesis are listed in Appendix A. Reaction buffer, template, primers, dNTP, and polymerase were mixed according to the manufacturer’s protocol, and PCR was performed. After PCR, 1 µL of *Dpn*I enzyme (20 U/µL) was added to PCR tubes, which were incubated at 37 °C for 1 h. Then, the solution was transformed into XL-10 Gold competent cells. The mutants were confirmed by DNA sequencing.

### 4.6. Gliotoxin Extraction and HPLC

*A. fumigatus* was grown in Czapek-Dox medium (gliotoxin-producing media/3% sucrose, 0.3% sodium nitrate, 0.05% magnesium sulfate, 0.05% potassium chloride, 0.1% potassium phosphate dibasic, 0.001% ferrous sulfate, final pH 7.3 ± 0.2 at 25 °C) [60] for 72 h at 37 °C in a shaking incubator. If necessary, zinc was added. After 72 h, gliotoxin was extracted using chloroform (1:1), as described previously [61]. The pellet was obtained through vacuum evaporation, resuspended in 99.9% pure methanol, and analyzed using RP-HPLC with a UV detector and a polar C18 RP-HPLC column (Agilent Eclipse XDB-C18 (5 µm) 4.6 × 250 mm). The flow rate was 1 mL/min, and the mobile phase was methanol:water (50:50).

### 4.7. Protein Purification

For ZafA protein preparation, the open reading frame of *ZafA* (1g10080) was generated by using a TOPscript™ cDNA synthesis kit (Enzynomics, Seoul, Korea) and then inserted into the pET21α expression vector to generate His-tagged ZafA protein by transformation into *E. coli* XL-10 Gold cells. *E. coli* transformants were incubated in a 37 °C shaking incubator. *E. coli* cells were cultured for 12–16 h at 18 °C and centrifuged at 3500 rpm, and the supernatant was removed. *E. coli* pellets were resuspended in 1 mL of lysis buffer (50 mM NaH_2_PO_4_, 300 mM NaCl, 10 mM imidazole, 1 mM PMSF, 0.5% NP-40, 1 mM DTT, 2 µg/mL lysozyme, 4 µg/mL leupeptin) and sonicated on ice for 10 s 3 times. The sonicated solution was centrifuged again, and the supernatant was collected. Ni-NTA (Ni^2+^-nitrilotriacetate-agarose resin (Qiagen, Seoul, Korea) was added to the supernatant and mixed gently by rocking at 4 °C for 1 h 30 min. Isolated Ni resin was washed 3 times with 500 µL of wash buffer (50 mM NaH_2_PO_4_, 300 mM NaCl, 20 mM imidazole). After removing the wash buffer, 100 µL of elution buffer (50 mM NaH_2_PO_4_, 300 mM NaCl, 250 mM imidazole) was added and incubated at 4 °C for 1 h. After 1 h, the supernatant was collected, and His-tagged ZafA protein was checked by the Bradford method using BSA as a standard.

### 4.8. ONPG Assay

A 608-bp fragment of the wild-type promoter region and mutated region of *GliZ* was digested with *Xho*I and *BamH*I and then ligated into pRS425 with *LacZ*. *ZafA* cDNA was cloned into pYPGE15. These two plasmids were cotransformed into BY4741 wild-type yeast. Cotransformed yeast cells were precultured in shaking SD-LU broth at 30 °C overnight. Precultured cells were inoculated in SD-LU broth without metals and cultured at 30 °C until the OD_600_ was 0.5–0.8. The yeast cells were pelleted and resuspended in Z-buffer (16.1 g/L Na_2_HPO_4_∙7H_2_O, 5.5 g/L NaH_2_PO_4_∙H_2_O, 0.75 g/L KCl, 0.246 g/L MgSO_4_∙7H_2_O, pH 7.2) and then lysed by chilling in liquid N_2_ and thawing at 37 °C in a water bath 3–5 times. Z-buffer containing β-mercaptoethanol and ONPG were subsequently added, and the solutions were incubated at 30 °C. The β-galactosidase units were calculated based on the OD_600_.

### 4.9. EMSA

The probe for the electrophoretic mobility shift assay (EMSA) was amplified from the genomic *GliZ* promoter region by PCR using the EMSA_F and EMSA_R primers. The probe was labeled by [^32^P] dCTP radioisotope. The EMSA experiment was performed as follows. The DNA-protein binding reaction was performed in 15 µL of binding buffer (1% (*w/v*) glycerol, 5 mM MgCl2, 50 mM KCl, 10 mM Tris-Cl (pH 8.0), 1 mM DTT, nuclease-free water) containing 1 µg of purified ZafA protein, ^32^P-labeled probes and unlabeled homologous competitor DNA or mutant competitors. This mixture was incubated for 30 min at room temperature. After 30 min of incubation, mixtures containing the DNA-protein complexes were electrophoresed in 4–15% native gels (Mini-PROTEAN^®^ TGX^TM^ Precast Gels, 4%–15%, 15-well comb, 15 µL/well, BIO-RAD, Seoul, Korea ) at 100 V. After electrophoresis, the gel was dried on 3M paper by a vacuum gel dryer and was then developed.

### 4.10. Conidial Killing Assay

HL-60 cells were cultured in RPMI 1640 medium containing 10% fetal bovine serum (FBS) and 1% Gibco™ Antibiotic-Antimycotic at 37 °C with 5% CO_2_. To induce the differentiation of HL-60 cells into macrophage-like and neutrophil-like cells, the HL-60 cells were stimulated with 1.25% dimethyl sulfoxide (DMSO) for 5 days (5) and 50 nM 12-o-tetradecanoylphorbol-13-acetate (PMA) for 2 days. We monitored cell differentiation over 24 to 48 h. Conidia obtained from the indicated fungal cells were incubated with RPMI 1640 medium at 37 °C for 4 h to obtain swollen conidia. A total of 10^3^ swollen conidia were transferred to 5 × 10^4^ differentiated HL-60 cells in RPMI 1640 medium supplemented with 5% FBS. After incubating for 0, 1, and 2 h, the sample tubes were vigorously agitated, and the number of surviving fungal cells was determined by quantitative culture. Swollen conidia in the absence of differentiated HL-60 cells were processed in parallel as a negative control. Differences between the groups were assessed using Student’s *t*-test for unpaired samples, and a *p*-value of less than 0.05 was considered significant.

### 4.11. qRT-PCR

Growth conditions of HL-60 cells and macrophage-like cell differentiation conditions were mentioned in the conidial killing assay (see materials and methods section). Here, 5 × 10^6^ macrophage-like cells were prepared per 90.00 × 15.00 (d × h (mm)) petri dish. Then, 1 × 10^7^ cells of *A. fumigatus* were added to the macrophages and incubated for 0, 1, or 2 h. Cells were washed with 1× PBS and harvested in 1× PBS. The samples were then centrifuged at 250× *g* for 5 min, and the supernatant was removed. Then, total RNA was extracted by using TRIzol^®^ (Invitrogen, Seoul, Korea), and cDNA was synthesized by using a TOPscript™ cDNA synthesis kit (Enzynomics, Seoul, Korea). qRT-PCR was performed by using KAPA SYBR^®^ for LightCycler^®^ 480.

### 4.12. RNA Sequencing

The total RNA extraction protocol from macrophages and *A. fumigatus* was described in the real-time PCR (see materials and methods section). RNA quality was assessed by an Agilent 2100 bioanalyzer using an RNA 6000 Nano Chip (Agilent Technologies, Amstelveen, the Netherlands), and RNA quantification was performed using an ND-2000 spectrophotometer (Thermo Inc., Delaware, USA). Libraries were prepared from 2 µg of total RNA using the SMARTer Stranded RNA-Seq kit (Clontech Laboratories, Inc., California, USA). Isolation of mRNA was performed using a poly(A) RNA selection kit (LEXOGEN, Inc., Wien, Austria). The isolated mRNAs were used for cDNA synthesis and shearing, following the manufacturer’s instructions. Indexing was performed using the Illumina indexes 1–12. An enrichment step was carried out using PCR. Subsequently, libraries were checked using the Agilent 2100 bioanalyzer (DNA High Sensitivity Kit) to evaluate the mean fragment size. Quantification was performed using the library quantification kit with the StepOne real-time PCR system (Life Technologies, Inc., California, USA). High-throughput sequencing was performed as paired-end 100 bp sequencing using a HiSeq 2500 (Illumina, Inc., California, USA). mRNA-seq reads were mapped using the TopHat software tool to obtain an alignment file. The alignment files were also used for assembling transcripts, estimating their abundances, and detecting differential expression of genes and isoforms using cufflinks. We used the fragments per kilobase of exon per million fragments (FPKM) as the method of determining the expression level of the gene regions. The FPKM data were normalized based on the quantile normalization method using EdgeR within R. The RNA-seq data have been deposited in NCBI’s Gene Expression Omnibus and are accessible through GEO series accession number GSE135818 (https://www.ncbi.nlm.nih.gov/geo/query/acc.cgi?acc=GSE135818). The network model was built using the Search Tool for Retrieval of Interacting Genes (STRING) App in Cytoscape software 3.7.0. The lists of the genes involved in the homeostasis of iron, copper, and zinc and the genes involved in gliotoxin biosynthesis from *A. fumigatus* genome were obtained from the web-based Gene Ontology browser (quickGO). The data analysis program we used was STRING:Protein Query and species was *A.fumigatus*. The confidence cutoff value was 0.4 and maximum additional interactors was 0. The expression level of proteins was represented by adding the log_2_ value to the fold change, comparing the values of mRNA expression level when zinc concentration was 0 and 1 uM. Proteins that did not show any interaction were not shown in the figure. The red dash line that we added represents the direct interaction of the ZafA protein to the *GliZ* gene.

### 4.13. Mouse Virulence Assay

Wild-type *A. fumigatus* and mutants were spotted on AMM agar plates and incubated at 37 °C for 5 days. Spores were harvested with a 0.9% NaCl solution containing 0.01% Tween 80 and filtered with Miracloth. The number of spores was counted with a hemocytometer (Marienfeld) to produce a concentration of 2.5 × 10^7^ spores/mL. BALB/c mice were purchased from Orientbio Co. The number of mice used in each experimental group was six, and all of the mice were six weeks old. They were immunosuppressed with cyclophosphamide (150 mg/kg body weight) on days –3, –1, +2 and every subsequent third day throughout the whole experimental period and with cortisone acetate (112.5 mg/kg body weight) on day –1. They were allowed free access to their food and 1 g/L tetracycline hydrochloride-supplemented drinking water. The mice were anesthetized with isoflurane, and 5 × 10^5^ spores (20 µL) were nasally infected per mouse. After the infection, the mouse survival rate was calculated.

### 4.14. Ethics Statement

All animal experiments were conducted in accordance with the Guidelines for the Care and Use of Laboratory Animals outlined by Ministry of Food and Drug Safety, Republic of Korea, and in accordance with protocols approved by Institutional Animal Care and Use Committee of Korea University (permit No. KUIACUC-2019-0105, 18 November 2019).

### 4.15. Statistical Analysis

All experiments including gliotoxin analysis, ONPG assay, conidial killing assay, and qRT-PCR were performed as triplicates. Mouse virulence assay was performed as two independent experiments. Differences between the groups were assessed using Student’s *t*-test for unpaired samples, and a *p*-value less than 0.05 was considered significant. Statistical significance was marked with asterisks. One asterisk (*) means that the *p*-value is less than 0.05, and two asterisks (**) less than 0.01.

## 5. Conclusions

The production of gliotoxin was regulated by ZafA, which is a zinc-dependent transcription activator. Two conserved ZafA-binding motifs, 5’-CAAGGT-3’, in the upstream region of GliZ gene influence to the gliotoxin production. These ZafA-binding motifs were also associated with virulence of Aspergillus fumigatus. Taken together, these results suggest that zinc plays an important role in the pathogenicity of A. fumigatus by regulating gliotoxin production during the phagocytosis pathway to overcome the host defense system.

## Figures and Tables

**Figure 1 ijms-20-06192-f001:**
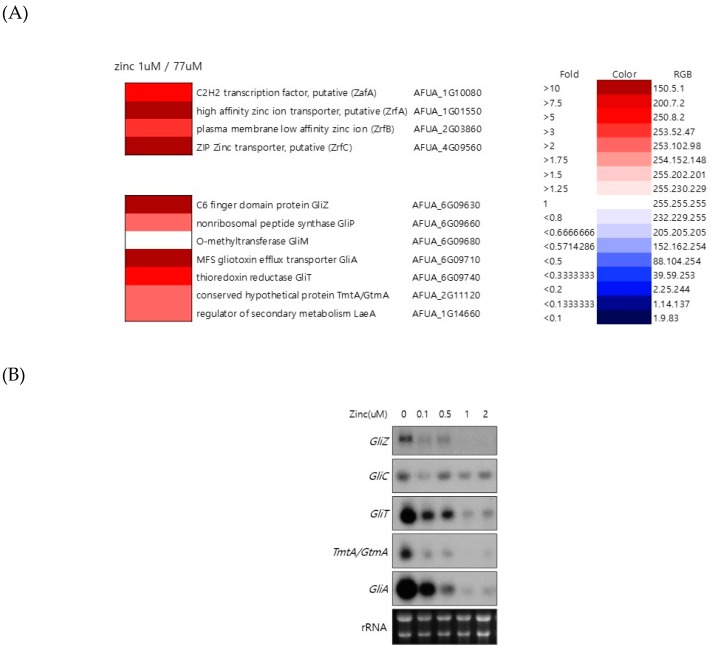
Zinc regulates the expression of genes involved in gliotoxin biosynthesis. (**A**) To investigate the genes regulated by zinc in *A. fumigatus*, microarray analysis was performed. Wild-type *A. fumigatus* was cultured in AMM with the indicated zinc concentration (1 and 77 µM), total RNA was extracted, and then microarray analysis was performed. The microarray data were clustered in zinc metabolism- and gliotoxin biosynthesis-related genes. (**B**) To confirm the microarray data, northern blot analysis was performed to detect the gene expression of *GliZ*, *GliT*, and *GliA*. rRNA was used as a loading control. The cells were cultured in AMM with the indicated concentration of zinc, total RNA was extracted, and then northern blot analysis was performed.

**Figure 2 ijms-20-06192-f002:**
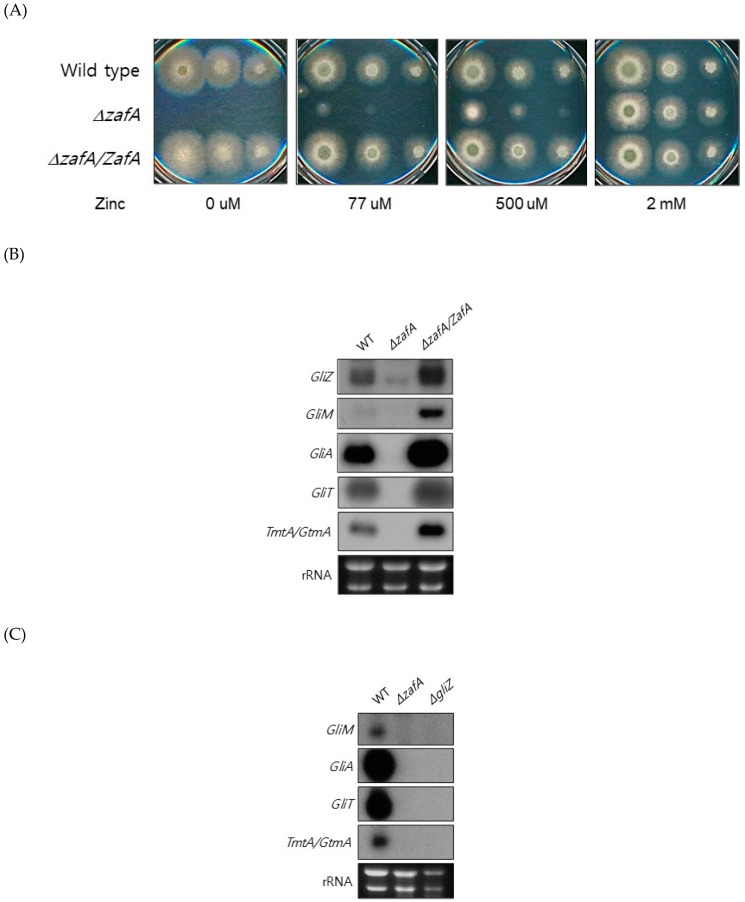
ZafA regulates gene expression of gliotoxin biosynthesis. (**A**) To investigate the effect of ZafA on the expression of gliotoxin biosynthesis genes, *ZafA* deletion mutant and *ZafA*-complemented *ZafA* deletion mutant strains were constructed as described in the materials and methods. The *ZafA* deletion mutant showed growth defects on low-zinc medium and AMM. Introduction of *ZafA* (*ZafA*-complemented strain) into the *ZafA* deletion mutant rescued the growth defect on low-zinc medium. (**B**) Northern blot analysis was performed to detect the gene expression of *GliZ, GliT*, and *GliA*. The indicated cells were cultured in PD broth, and total RNA was extracted. (**C**) Northern blot analysis was performed with the *GliZ* deletion mutant to detect the gene expression of *GliT* and *GliA*. The indicated strains were cultured in PD broth, and the expression of *GliA* and *GliT* was downregulated in both the *GliZ* and *ZafA* deletion mutants.

**Figure 3 ijms-20-06192-f003:**
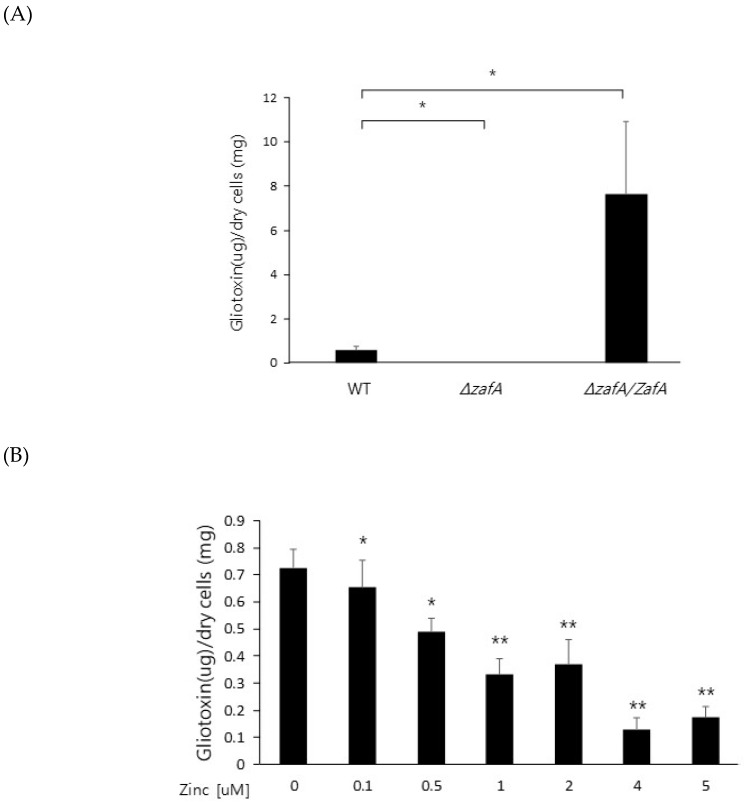
Zinc downregulates gliotoxin production. (**A**) To investigate gliotoxin production in the *ZafA* deletion mutant, the indicated strains were cultured in Czapek-Dox medium for 3 days at 37 °C, and gliotoxin was extracted from the culture medium. The produced gliotoxin was measured with HPLC. (**B**) The effect of zinc on gliotoxin production was investigated. Wild-type cells of *A. fumigatus* were cultured in Czapek–Dox medium with the indicated concentration of zinc for 3 days at 37 °C, and gliotoxin was extracted from the culture medium. The produced gliotoxin was measured with HPLC. (**C**) The *GliZ* deletion mutant was used as a control for gliotoxin production to identify the function of GliZ in gliotoxin production. * *p* < 0.05, ** *p* < 0.01.

**Figure 4 ijms-20-06192-f004:**
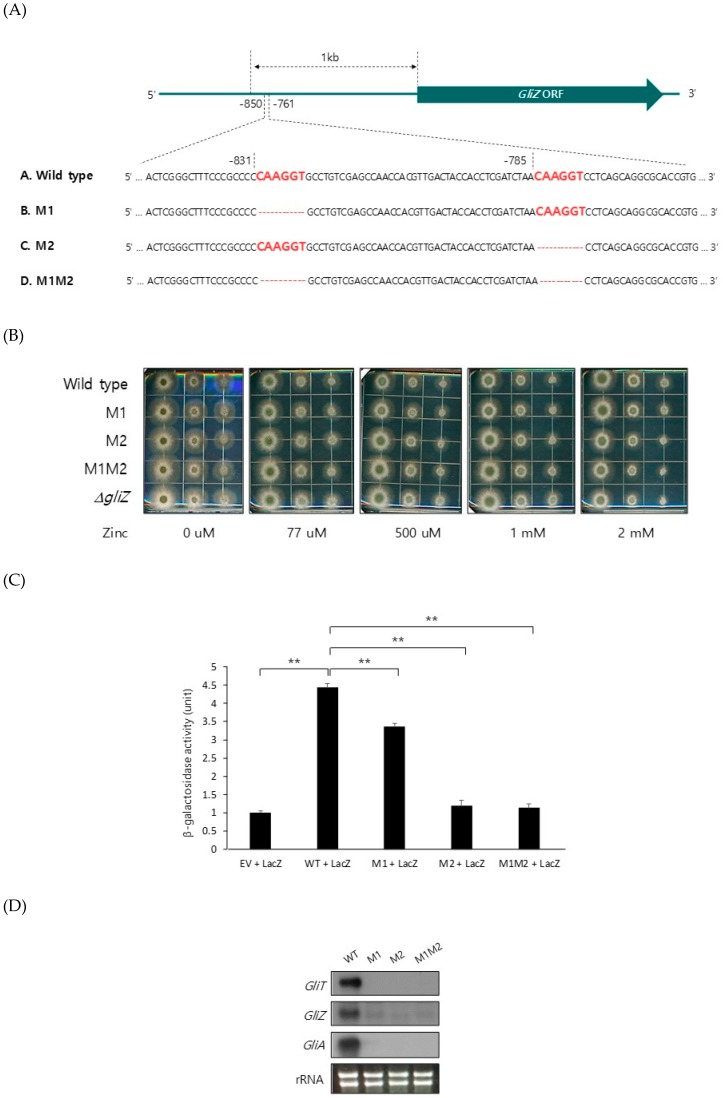
ZafA binds specifically to the upstream region of *GliZ* at conserved ZafA-binding motifs and regulates *GliZ* gene expression. (**A**) Two conserved ZafA-binding motifs, 5′-CAAGGT-3′, were detected in the upstream region of *GliZ*, and a deletion of the ZafA-binding motif was constructed at the −831 region (M1), −785 region (M2), and −831 and −785 regions (M1M2). The growth of the M1, M2, and M1M2 mutants was not different from that of wild-type cells on AMM (**B**). The conidia of the indicated strains were diluted 10-fold, and diluted conidia were spotted on AMM and incubated for 2 days at 37 °C. (**C**) To investigate the direct interaction between ZafA and the *GliZ* promoter region, an ONPG assay was performed in *S. cerevisiae*. PCR-amplified 608-bp fragments of the wild-type promoter and mutated promoters (M1, M2, and M1M2) were subcloned into a yeast vector with the beta-galactosidase gene and cotransformed into yeast cells with a *ZafA* expression vector. Then, the cotransformants were cultured in SD media, and the ONPG assay was performed as described in the materials and methods section. An empty vector was used as a control. (**D**) Furthermore, northern blot analysis was performed to investigate the effect of mutation of the ZafA-binding motifs of the *GliZ* upstream region on the expression of *GliZ*, *GliT*, and *GliA*. The indicated strains were cultured in AMM until the exponential phase, and total RNA was extracted. (**E**) To confirm the binding of ZafA to the ZafA-binding motifs of the *GliZ* upstream region, an EMSA was performed. First, 123-bp fragments of the promoter regions of *GliZ* from the wild-type and indicated mutant strains were amplified by PCR, and DNA sequences were confirmed. ZafA protein and the ^32^P-labeled wild-type *GliZ* promoter were incubated with the wild-type *GliZ* promoter or each mutant promoter (M1, M2, and M1M2) as competitors. Then, the reaction mixtures were separated using native PAGE. M1, M2, and M1M2 indicate the mutated promoters of *GliZ*. Competitors were added at 10-fold or 350-fold excess to the labeled probe. (**F**) Gliotoxin production was measured. The wild-type and each mutant strain were cultured in Czapek-Dox medium for 3 days at 37 °C, and gliotoxin was extracted from the culture medium. The produced gliotoxin was measured with HPLC. The *GliZ* deletion mutant was used as a control. The graph shows the relative amount of gliotoxin produced by the indicated strains. * *p* < 0.05, ** *p* < 0.01.

**Figure 5 ijms-20-06192-f005:**
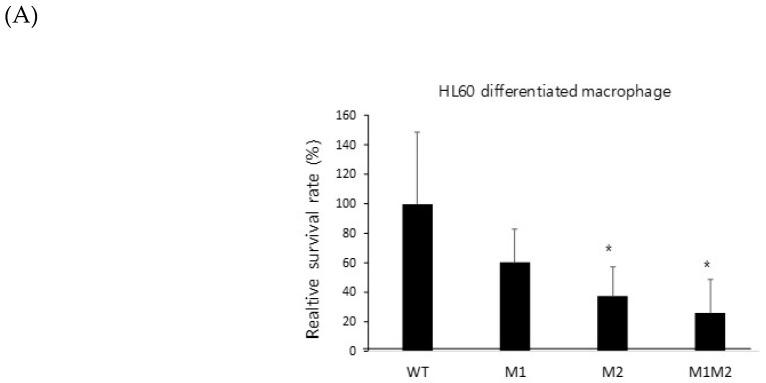
Mutation of the ZafA-binding motif decreased the virulence of *A. fumigatus*. (**A**) A conidial killing assay with HL-60 differentiated macrophage cells was performed. A total of 1 × 10^6^ conidia of *A. fumigatus* were added for 1 h, and the infected cells were washed with 1× PBS and harvested with 1× PBS. Cells were resuspended in 100 µL of 1× PBS and lysed, and then cell lysates were incubated on AMM. The same experiments were performed with the HL-60 differentiated neutrophil cells (**B**). The graph shows the relative number of fungal colonies grown on AMM compared to that of wild-type cells. (**C**) A virulence test was performed with a murine model, and 5 × 10^5^ conidia of the indicated strains were used to infect mice. The mice were anesthetized with isoflurane, and 5 × 10^5^ spores (50 µL) were nasally infected per mouse. The number of mice used in each experimental group was six, and all of the mice were six weeks old. Curves were plotted by using GraphPad Prism software. The *p*-value was 0.0006. (**D**) A fungal burden assay was performed. The mice infected with conidia of the wild-type or mutant strains were sacrificed, and the mouse lungs were dissected after 3 days of infection, fixed in 10% formalin, and then embedded in paraffin. Hematoxylin and eosin (H&E) and Grocott methenamine silver (GMS) staining were performed. NC and PC indicate negative and positive controls, respectively. GMS staining quantification was performed using “Image J” program. We transformed images into RGB stack files, and their GMS % area was measured as signals respectively. The samples including wild type, M1M2, and *gliZ* mutants were compared respectively based on the signal of negative control (= 1). * *p* < 0.05, ** *p* < 0.01.

**Figure 6 ijms-20-06192-f006:**
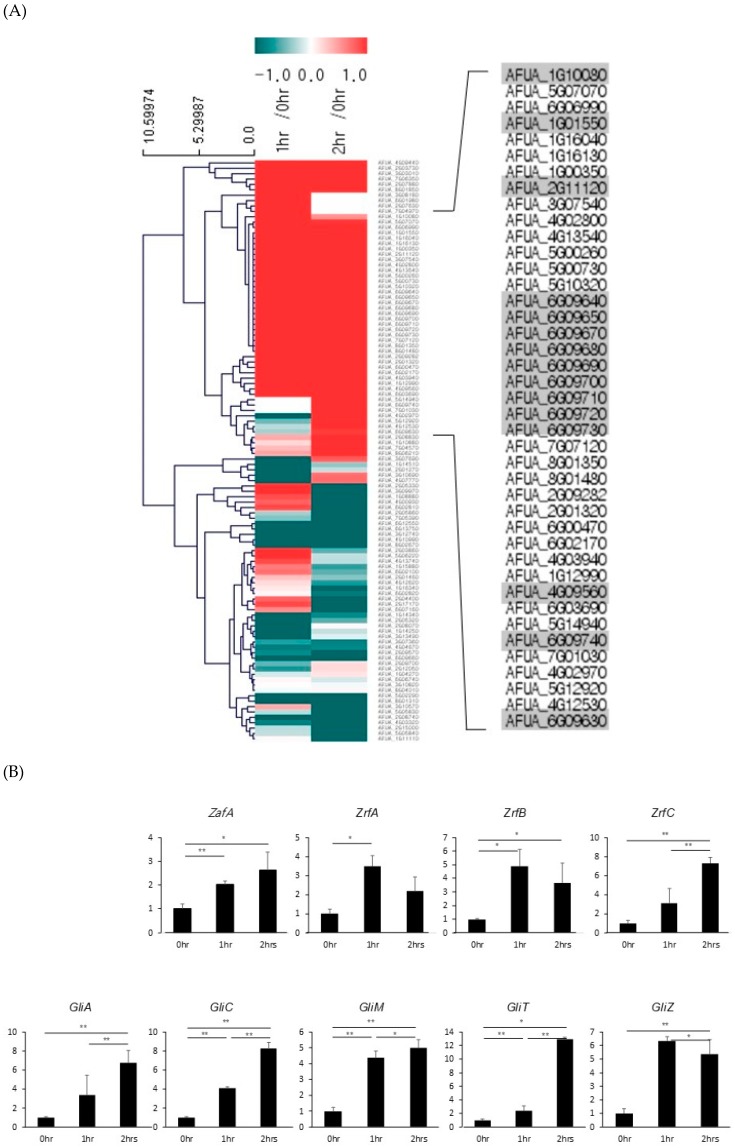
Zinc-related and *gli* cluster genes were upregulated during phagocytosis. (**A**) Total RNA was extracted from the macrophage phagocytic cells infected with conidia of *A. fumigatus*, and libraries were prepared from 2 µg of total RNA. High-throughput sequencing was performed as paired-end 100 bp sequencing using a HiSeq 2500 (Illumina, Inc., USA). mRNA-seq reads were mapped using the TopHat software tool to obtain an alignment file. (**B**) Total RNA was extracted from the phagocytic cells infected with conidia after 0 and 1 h postinfection, and cDNA was synthesized using a TOPscript™ cDNA synthesis kit (Enzynomics). Then, qRT-PCR was performed using KAPA SYBR^®^ for LightCycler^®^ 480 with the primer sets for the indicated genes. (**C**) The transcriptomic profile during phagocytosis is shown. The gene expression profile was grouped into zinc, copper, iron, and gliotoxin metabolism. Upregulated genes were found in all categories, and *ZafA* and *GliZ* were upregulated during phagocytosis. Up- and downregulated genes are colored to show fold change. * *p* < 0.05, ** *p* < 0.01.

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
