# Peer review of "The Role of Zinc in Gliotoxin Biosynthesis of Aspergillus fumigatus"

_ijms, 2019, doi:10.3390/ijms20246192_

Round 1

Reviewer 1 Report

The reviewers comments have addressed. 

Author Response

Thank you for your review and I appreciate it.

Reviewer 2 Report

The manuscript entitled “The role of zinc in gliotoxin biosynthesis of Aspergillus fumigatus” contains a lot of interesting and important data about the roles of zinc in gliotoxin production and virulence. However, to accept this Journal, it need to revise some aspects.

Major

The statistical analysis and ethic statement section should be needed. The statistical significances were not appeared in all data containing mouse survival curve. Fig. 2B, Line 319; Why complement strain showed higher GliZ, GliT, and GliA transcripts, and gliotoxin production than WT? The authors address these phenomena in depth. How many repetition of RNA-seq? Please describe more detail.

Minor

Line 81-82; “the production of GT is regulated by zinc availability” was not described in Reference 34. Please correct it. Fig. 1; Why authors detected only GliZ, GliT, and GliA transcripts? The northern data for GliP, GliM, TmtA, LaeA should be needed. Also, Fig. 1B, the data of rRNA showed different origin. Please check it. Line 426; The fungal burden assay was not quantified. Please add quantification data. Line 470; “gli” to “gli

Author Response

The statistical analysis and ethic statement section should be needed.

- We added statistical analysis and ethic statement in materials and methods.

The statistical significances were not appeared in all data containing mouse survival curve.

- We made all figures again as reviewer commented and statistical significances and methods are added in materials and methods. We performed mouse survival experiments as two independent experiments and showed same pattern with the figure. However, the date of death is little bit different each other, and it is difficult to make graph when we made graph with two experiments. If you want, we can show another survival data. 

Fig. 2B, Line 319; Why complement strain showed higher GliZ, GliT, and GliA transcripts, and gliotoxin production than WT? The authors address these phenomena in depth.

- Well, this is very difficult question. But when we perform same experiments, we observed same results. Interestingly, transformants showed higher activity than wild type in fungal and yeast model system. One possibility is that the transformants have pyrG selection marker and it may activate expression level than wild type. 

How many repetition of RNA-seq? Please describe more detail.

- We performed three or four times. The statistical significances were shown in Figure 6.

Minor

Line 81-82; “the production of GT is regulated by zinc availability” was not described in Reference 34. Please correct it.

- We corrected it.

Fig. 1; Why authors detected only GliZ, GliT, and GliA transcripts? The northern data for GliP, GliM, TmtA, LaeA should be needed.

- We added all data except LaeA and GliP. They were not detected from northern blotting although we tried several times.

Also, Fig. 1B, the data of rRNA showed different origin.

- When we performed northern blotting, we did not carry out re-probing because of intensity. So each blot has individual rRNA, and the rRNA level was same in all experiments.

The fungal burden assay was not quantified. Please add quantification data.

- We added quantification data in Figure 5D.

Line 470; “gli” to “gli” 

- We changed as you commented.

Round 2

Reviewer 2 Report

Now the manuscript is acceptable to publication with some minor revisions.

Please define the gene name and protein name (ex gliT vs GliT). Confirm P-value or p-value. Why don’t you use “Prism” program for survival curve? “Well, this is very difficult question. But when we perform same experiments, we observed same results. Interestingly, transformants showed higher activity than wild type in fungal and yeast model system. One possibility is that the transformants have pyrG selection marker and it may activate expression level than wild type. “  Would you cite and discuss those cases in suitable section.

Author Response

Please define the gene name and protein name (ex gliT vs GliT). Confirm P-value or p-value.

- We changed all gene names as reviewer commented. (GliT, P-value)

Why don’t you use “Prism” program for survival curve?

- We made figure again as reviewer commented using “Prism” software.

Well, this is very difficult question. But when we perform same experiments, we observed same results. Interestingly, transformants showed higher activity than wild type in fungal and yeast model system. One possibility is that the transformants have pyrG selection marker and it may activate expression level than wild type. “Would you cite and discuss those cases in suitable section. 

- We added the explanation about it in the result section. (lines 340-343)

This manuscript is a resubmission of an earlier submission. The following is a list of the peer review reports and author responses from that submission.

Round 1

Reviewer 1 Report

ijms-621443

Seo et al. The role of zinc in gliotoxin biosynthesis of Aspergillus fumigatus

The manuscript is well written, it is well structured and experiments carefully performed and analysed. The findings are important and well supported by the experiments.

Aspergillus fumigatus is an important pathogen and gliotoxin the toxic compound produced by this mold. The authors identified the transcription regulation for the proteins that produce this compound. This is novel and of medical relevance. This might be an important step to fight aspergillosis.

minor: line 85 even?

Author Response

Aspergillus fumigatus is an important pathogen and gliotoxin the toxic compound produced by this mold. The authors identified the transcription regulation for the proteins that produce this compound. This is novel and of medical relevance. This might be an important step to fight aspergillosis.

minor: line 85 even?

> Thank you for your kind comments and as you commented, line 85 was changed to "even though". Thank you again.

Reviewer 2 Report

The article presents a study about the role of zinc in gliotoxin biosynthesis. The researchers used A. fumigatus, a known pathogenic fungus, to investigate the correlation between zinc starvation and gliotoxin formation and then they verified how relevant this interaction is during virulence.

The correlation between gliotoxin biosynthesis and the zinc-responsive transcriptional activator ZafA was already shown in another article. However, in this paper, the authors were able to better determine the ZafA DNA-binding sites on the gliZ promoter region. Indeed, mutagenesis of those binding motifs resulted in decrease gliZ activation and consequentially in a repression of the gliotoxin gene cluster. As a further proof, the mutant strains presented also decreased virulence in a murine infection model similar to the gliZ mutant strain.

There are some parts of this manuscript that are novel and of interest for the community, but other data are redundantly reported. The first part of the manuscript, namely the results reported in figures 1, 2, and 3 are not new and they all cover the same aspect: the importance of zinc starvation for gliotoxin production. Afterward, the authors focused on the ZafA DNA-binding sites on the gliZ promoter, which is the novel part, and then, in figure 6 a network modeling that is not much related to the rest of the article. The feeling is that more than half of the data presented here have been shown only to increase the mole of results without a rational design.

Major concerns:

The article needs to be strongly edited. Too many concepts are redundantly repeated. If you check the first 12 lines of the introduction the authors used the term “living cells” 6 times and the term “living organism” 3 times. As another example, the sentence “zinc regulates gliotoxin biosynthesis” was repeated at list 6 times in the entire manuscript. Additionally, the references are not formatted. The network modeling is not explained at all. Data are missing concerning: the used mathematical model; the mRNA-seq experiment; where the data are deposited. As it is, this part can be completely ignored.

Minor concerns:

Table 1 and 2 can go in the supplementing.

Author Response

Thank you for your valuable comments and it is a help to prepare my manuscript. Followings are answers the questions. 

Major concerns:

The article needs to be strongly edited. Too many concepts are redundantly repeated. If you check the first 12 lines of the introduction the authors used the term “living cells” 6 times and the term “living organism” 3 times. As another example, the sentence “zinc regulates gliotoxin biosynthesis” was repeated at list 6 times in the entire manuscript.

> As you commented, same phrase was repeated. So we changed or deleted many phrase as you commented.

Additionally, the references are not formatted.

> References were formatted to fit the journal.

The network modeling is not explained at all.

> A description of network modeling has been added in result part.

Data are missing concerning: the used mathematical model; the mRNA-seq experiment; where the data are deposited. As it is, this part can be completely ignored.

> The RNA-seq data have been deposited in NCBIs Gene Expression Omnibus and are accessible through GEO series accession number GSE135818 (https://www.ncbi.nlm.nih.gov/geo/query/acc.cgi?acc=GSE135818).

Minor concerns:

Table 1 and 2 can go in the supplementing

> Table 1 and 2 are moved to supplementary data part.

Your comments were helpful and I appreciate it.

Round 2

Reviewer 2 Report

I have checked the resubmitted version of the article and I noticed that the authors did not put many efforts to improve the quality of the manuscript. I suggested a strong editing of the manuscript, but the authors just deleted some of the redundant phrases. I asked to explain the mathematical model they have used for the network, and they claimed that “A description of network modeling has been added in result part”, which is not true. I have seen that they submitted the data in the GEO system, which is at least something.

The article was resubmitted almost in the same shape as before and I found that a bit disrespectful. In my opinion, as presented, this article cannot be published.